# Beliefs and Sociodemographic and Occupational Factors Associated with Vaccine Hesitancy among Health Workers

**DOI:** 10.3390/vaccines10122013

**Published:** 2022-11-25

**Authors:** Tânia Maria de Araújo, Fernanda de Oliveira Souza, Paloma de Sousa Pinho, Guilherme Loureiro Werneck

**Affiliations:** 1Health Department, State University of Feira de Santana, Feira de Santana 44.036-900, Brazil; 2Health Sciences Center, Federal University of Recôncavo da Bahia, Santo Antônio de Jesus 44.430-622, Brazil; 3Department of Epidemiology, Institute of Social Medicine, State University of Rio de Janeiro, Rio de Janeiro 20950-000, Brazil

**Keywords:** vaccination, healthcare workers, vaccine hesitancy, vaccination delay, vaccination awareness, attitude to health, community health workers, health personnel

## Abstract

**Introduction**: Vaccine hesitancy has been implicated in the low-vaccination coverage in several countries. Knowledge about vaccine hesitancy predictors in health workers is essential because they play a central role in communication about the importance and safety of vaccines. This study aimed to assess beliefs and sociodemographic and occupational factors associated with vaccine hesitancy in health workers. **Methods**: This was a cross-sectional study among 453 health workers in primary and medium complexity services in a municipality in the state of Bahia, Brazil. The variable vaccine hesitancy was operationalized based on the answers related to incomplete vaccination against hepatitis B, measles, mumps and rubella, and diphtheria and tetanus. Associations between variables were expressed as prevalence ratios (PR) and their respective 95% confidence intervals (CI). **Results**: Endemic disease combat agents, administrative service workers, and support staff had the highest levels of vaccine hesitancy. Among the analyzed variables, the following were associated with vaccine hesitancy: working in secondary health care services (PR: 1.21; CI: 1.07–1.36), working as an endemic disease combat agent (PR = 1.42; 95% CI: 1.165–1.75), not sharing information about vaccines on social media (PR = 1.16; 95% CI: 1.05–1.28), distrusting information about vaccinations (PR: 0.86; CI: 0.75–0.99), and not feeling safe receiving new vaccines (PR = 1.16; 95% CI: 1.06–1.28). **Conclusions**: Strategies to enhance confidence in vaccination among health workers should consider differences in occupations and their working settings. Improving vaccination-related content in training and continuing education activities and facilitating access to onsite vaccinations at the workplace are crucial elements to reduce vaccine hesitancy among health workers.

## 1. Introduction

At the end of the 20th century, the Center for Disease Control and Prevention (USA) published a list of the greatest achievements in public health [1]. Vaccinations had a prominent position in the list because they alone are responsible for increasing life expectancy by approximately 30 years. It is known that vaccination is the most effective way of preventing a wide range of diseases. Furthermore, it indirectly decreases the risk of those who remain susceptible in the community in general, reducing the burden of morbidity and mortality associated with communicable diseases [2].

In spite of the recognized importance of vaccines, different countries face problems related to a lack of trust in vaccinations, which has been implicated in low-vaccine coverage in different groups, such as children [3], adolescents [4], adults [5], pregnant women [6], older adults [7], and health workers [8,9,10,11,12]. A lack of trust leading to vaccine hesitancy has become more evident recently, with the development of vaccines against COVID-19. Vaccine hesitancy is defined as any delay in the acceptance or refusal of vaccination despite the availability of vaccination services [13]. Even in the context of the pandemic, vaccine hesitancy among adults has been frequently registered due to misinformation about the efficacy and safety of vaccines, reduced health literacy, organizational constraints, and reduced feelings of social responsibility [14,15].

In Brazil, the vaccination schedule for health workers recommended by the Ministry of Health includes vaccines against hepatitis B, diphtheria and tetanus (dT), and measles, mumps, and rubella (MMR) [16]. Despite these recommendations and the free-of-charge access to vaccines, a complete vaccination schedule as low as 38.5% has been described among health workers [12]. Among the possible challenges to reaching high-vaccination rates in Brazilian health workers is the availability of vaccination rooms near their workplaces. Since vaccination rooms are mainly located in primary care units, workers outside these settings are likely to have fewer opportunities for vaccination. In addition, most healthcare services have no standardized reminder system for individual vaccination assessment. That is, there is no systematic monitoring or surveillance of the vaccination status of health workers.

In view of frequent exposures to infections in the work environment, deriving from the assistance health professionals provide to people with infectious diseases, handling of sharp materials, and direct contact with secretions, blood, and organic fluids, protective measures are necessary to minimize the possibility of infection [17]. One of the most important preventive measures available is vaccination, being a crucial measure, especially among these workers.

The literature shows that, similar to the general population, health workers can be hesitant about receiving the recommended vaccines [18,19,20]. This is a worrying situation, as health workers, in addition to the increased risk of contracting vaccine-preventable diseases, can also transmit them to the individuals they assist or live with.

Health workers are considered the cornerstone in maintaining confidence in vaccination [21] because they are directly responsible for vaccination activities, particularly in explaining the benefits and risks of vaccines. Together with their knowledge, workers’ attitudes and practices determine how they recommend a vaccine to the people they serve [22].

Considered a complex behavioral phenomenon in regard to its determinants, vaccine hesitancy can be described on a continuum that ranges from the positive extreme—those who accept all vaccines with no doubts—to the negative extreme—those who refuse everything with no doubts. The review of these models reinforced that vaccine hesitancy is complex and is not driven by a simple set of factors. Aspects such as complacency, convenience and trust should be considered [13,23].

The heterogeneous group of individuals between the two extremes exhibit various degrees of ‘hesitancy’. The study of this phenomenon reflects a shift in the focus of research, which was mainly concerned with promoting and guaranteeing the supply and availability of vaccines, and now seeks to explore the factors that interfere in people’s willingness to accept vaccination for themselves or their children when supply and access are guaranteed [24].

Larson et al. [25] highlight some points that should be considered in the study of vaccine hesitancy. The first one is that refusal alone is not the same as hesitancy. That is, counting only vaccine refusers will not capture the dimension of hesitant individuals [13]. Due to this, it is essential to assess the different factors that influence people not to vaccinate themselves or not to receive all the recommended vaccines. Second, it should be considered that vaccine hesitancy might be specific to some vaccines, but not all. Therefore, interpretation of surveys should be cautious, unless responses in relation to a range of vaccines are stated [25].

Studies that have examined vaccination-related barriers are often limited to hospital service professionals [8]. Thus, understanding this phenomenon concerning hesitation about vaccination among health workers in the most diverse workplaces (such as in primary care services) may guide actions to expand vaccinations [26]. The focus on health workers is justified because these professionals play a crucial role in promoting vaccine acceptance in addition to being at an increased risk of infection [12].

The majority of papers investigating vaccine hesitancy among health workers focus on influenza vaccines, followed by vaccines against hepatitis B, whooping cough, smallpox, HPV, and varicella. Few studies approach vaccine status in general [26]. There is a gap in knowledge about vaccine hesitancy in different categories of workers, especially concerning health workers who play a central role in communication about the importance and safety of vaccines [18], such as the health agents who carry out home visits and monitor the health situation of populations.

In spite of their central role in maintaining the public’s confidence in the vaccine acceptance process, health workers are also exposed to situations and factors that determine their own vaccine hesitancy. Therefore, this study aimed to assess beliefs and sociodemographic and occupational factors associated with vaccine hesitancy in health workers.

## 2. Material and Methods

This is a cross-sectional, exploratory study that is part of a large research “*Vigilância e Monitoramento de Doenças Infecciosas entre Trabalhadores e Trabalhadoras do Setor Saúde*” (Surveillance and Monitoring of Infectious Diseases among Health Workers), carried out by the Epidemiology Center of the Feira de Santana State University (UEFS). The primary purpose of the research program is to develop knowledge and social technologies to structure public policies for improving the health and working conditions of health workers in Bahia, Brazil. Specifically, we aim to estimate the prevalence of infectious diseases and vaccine uptake among health workers, and to evaluate preventive measures and propose a long-term monitoring system for health workers.

### Participants and Samples

This survey included workers in effective professional practice who worked in primary care and medium complexity services in urban and rural areas of a municipality in Bahia, Northeast Brazil, in 2019. Professionals in primary care services mainly work in clinical care, surveillance, and health promotion activities. Among them are the so-called “community health agents” and the “endemic disease combat agents”. The first develops health promotion and disease preventive actions, focusing on health education activities in households and communities. The second, also works outside the health unit inspecting houses, deposits, vacant lots, and commercial establishments with the aim of preventing and controlling infectious diseases.

Medium-complexity health services are not directly related to the vaccination procedures and include health workers from different backgrounds developing a variety of activities such as specialized clinical and psychosocial care, emergency care, occupational health monitoring, testing and counseling for infectious diseases, and health management [12].

All the workers who were effectively working and enrolled in the National Register of Health Facilities (CNES), were eligible for the study. Nominal lists of these workers, provided by the Municipal Health Department, were checked at the workplace during the initial visits to the health services.

To calculate the sample size, we considered the total population of professionals working in the services (622), a 61.5% prevalence of vaccine hesitancy [27], a 3% margin of error, and a 95% level of confidence. We estimated a sample of 380 workers to investigate the outcome of interest. As the study is part of a broader study that investigated other health outcomes, its sample size was larger than the one established for the analysis of the vaccine situation (N = 453).

The study population was selected by stratified random sampling taking into account the level of care of the services and occupational group, and was based on a previous survey of the health network’s structure and of the municipal service’s workforce. The selection was performed out of a list containing all the health workers of the services, considering those who were eligible for the study. Further selection was performed by a random number list generated by the Statistical Package for Social Sciences (SPSS), version 22.0 (IBM Corp, Armonk, NY, USA), considering the strata of levels of care (primary and secondary care) and occupational groups.

## 3. Data Collection and Study Variables

Data were collected using two procedures. For professionals with higher education degrees, we used a structured, self-administered questionnaire. Considering possible language barriers and different levels of education, a face-to-face questionnaire was additionally administered by trained interviewers for professionals with lower schooling levels. A pilot study was conducted to test and standardize procedures to minimize information bias.

The multidimensional questionnaire used for data collection included questions built specifically for this research, based on a literature review focusing on work and health conditions in the healthcare sector, and questions concerning vaccination-related beliefs, extracted from a validated questionnaire [28]. The instrument consisted of eight blocks of questions. The present study analyzed the following blocks: I—general identification; II—occupational characteristics; III—life habits and health-related aspects, including vaccination profiles and vaccination influences, attitudes, and experiences.

The outcome variable was vaccine hesitancy. The analysis of this outcome included an assessment of vaccine status, considering the vaccination schedule for three vaccines advocated by the National Immunization Program (PNI): diphtheria and tetanus (vaccinated in the last 10 years = 0; no = 1), hepatitis B (three doses = 0; two doses, one dose, or no dose = 1), and measles, mumps, and rubella (two doses = 0; one dose or no dose = 1). The variable vaccine hesitancy was operationalized based on the sum of the answers to these questions. The workers who answered yes in all the questions (score totaling 0) were considered the reference category (nonhesitant); those workers whose score was ≥1 were considered hesitant.

The independent variables included in the study referred to socioeconomic characteristics (sex, age, level of schooling, race or skin color, and income), occupational characteristics (working day, employment relationship, occupation, length of time in the profession, and level of care), and vaccination influences, attitudes, and experiences (injection fear, adverse reactions, vaccine confidence, media influence, distance of vaccination units, participation in campaigns, perceptions of risk and benefits, financial impediments to vaccination, and the way the person was treated (reception) in vaccination services—reception conditions).

The workers were categorized by occupational group: community health agents, endemic disease combat agents, health professionals with higher education degrees (physicians, nurses, dentists, physiotherapists, social workers, nutritionists, pharmacists, psychologists, and occupational therapists), health technicians (dental, nurse, and laboratory technicians), administrative services and support staff (reception, general services, doormen, security personnel). Prevalence of complete vaccination schedule, incomplete vaccination schedule, and nonvaccination for at least one immunizer was analyzed according to the occupational categories.

## 4. Statistical Analysis

In the bivariate analysis, prevalence ratios (PR) and their respective 95% confidence intervals (95% CI) were calculated. To assess the measurement of statistical significance, Pearson’s chi-square test was used, with *p* < 0.05.

The multivariate analysis aimed to describe the simultaneous effect of the variables of interest on vaccine hesitancy. To accomplish this, some procedures were followed: selection of variables based on the literature review and on the study’s objective; verification of the model’s assumptions; and preselection of variables considering *p* ≤ 0.25 in the bivariate analysis [29]. Finally, the adjusted analysis included all preselected variables, and the estimates of association and fit of the model were evaluated and estimates of association and adjustment of the model were assessed with and without the variable under investigation. In this multivariate analysis, a logistic regression model was used. PR estimates were obtained using point estimates, standard errors of the delta method, and respective 95% confidence intervals, calculated using the logit post estimation command in STATA (adjrr) [30].

It is important to mention that the variables “injection fear” and “history of reactions” were included in the multivariate models due to the theoretical consistency of their influence on the outcome [19,26,31,32]. The models were considered well-adjusted and without multicollinearity between the adjustment variables.

The diagnosis of the final model was assessed by the goodness-of-fit test [29] and by the area under the ROC curve. The results of the techniques were compared and the models with the lowest values of the Akaike information criterion (AIC) were selected.

The data were input using the SPSS version 19.0 (IBM Corp, Armonk, NY, USA) and STATA version 13.0 (Stata Corp LP, College Station, TX, USA).

### Ethical Aspects

The study followed the criteria established in the ethical principles contained in resolution no. 466/12 of the National Health Council. It had been previously approved by the Ethics Committee of the Feira de Santana State University (registration no. 2.897.062).

## 5. Results

A total of 453 workers from the healthcare network were investigated, of whom 352 (77.7%) worked in primary care and 101 (22.3%) in medium complexity services. These workers were characterized by a higher proportion of women (82.8%), aged 40 years or older (55.2%), with brown skin color (50.4%), earning up to two minimum wages (67.7%), with a permanent employment relationship (69.1%), and some perception of exposure to biological material (11.8% rarely; 20.5% sometimes; 22.8% always) (Table 1).

Among the investigated health workers, 96.2% did not think that distance to clinics or units prevented vaccination and 89.8% stated that the lack of financial means never prevented them from being vaccinated, but 29.7% reported having a history of reactions concerning some vaccine (Table 2).

A total of 27.8% of the health workers did not receive the three doses of the hepatitis B vaccine, 40.4% did not complete the immunization schedule for the triple viral vaccine (measles, mumps, and rubella) proposed for health workers, and 39.0% reported an incomplete immunization schedule concerning the diphtheria and tetanus vaccine (Table 3).

Figure 1 presents the prevalence of vaccine hesitancy for the three investigated vaccines across the different occupational categories. The highest levels of vaccine hesitancy were found in endemic disease combat agents (60.9%), followed by administrative services workers and support staff (36.6%).

Considering the outcome vaccine hesitancy, we observed significant and positive associations among the workers in general for men (PR: 1.18 CI: 1.07–1.29), working in medium complexity care services (PR: 1.15; CI: 1.05–1.26), working as endemic disease combat agents (PR:1.46; CI: 1.20–1.78), working in administrative and support services (PR: 1.32; CI: 1.08–1.62), not sharing news about vaccination on the social media (PR: 1.23; CI: 1.12–1.34), not believing that the government offers the best vaccine (PR:1.11; CI: 1.01–1.22), not feeling safe to receive new vaccines (PR: 1.19; CI: 1.09–1.30), distance to vaccination clinics or units (PR: 1.18; CI: 1.03–1.36), not having enough information to decide on being vaccinated (PR: 1.13; CI: 1.02–1.24), not having the financial means to be vaccinated (PR: 1.13; CI: 1.01–1.27), and not considering the vaccination schedule to be flexible (PR: 1.22; CI: 1.13–1.33) (Table 4).

In the multivariate analysis, the following remained associated with vaccine hesitancy: working in medium complexity care services (PR: 1.21; CI: 1.07–1.36), working as endemic disease combat agents (PR: 1.42; CI: 1.165–1.75), not sharing news about vaccination on social media (PR: 1.16; CI: 1.05–1.28), not trusting information about vaccination (PR: 0.86; CI: 0.75–0.99), and not feeling safe to receive new vaccines (PR: 1.16; CI: 1.06–1.28) (Table 5).

## 6. Discussion

The main findings show a wide variation in vaccine coverage for the different vaccines that are recommended for all the workers of SUS (Brazilian National Health System). The immunization rates of professionals were far from reaching the 90% coverage recommended by PNI. The great majority of workers were in the range related to the group of hesitant individuals—with some delay in the proposed vaccination schedules—despite the availability of vaccines. It is worth highlighting that the workers who did not hold a formal degree in health courses reported a lower rate of complete schedule for all the vaccines, as well as low perception of exposure to biological materials.

It is known that the perception of susceptibility to diseases enhances the perception that vaccination is important for health maintenance; therefore, it is a determinant factor for accepting vaccines. Such perception varies according to the feelings of personal vulnerability to a certain exposure; thus, it is related to an individual’s subjective perception of the risk of contracting diseases. It varies between individuals with some people who deny any possibility of contracting a disease, some that admit that there is a possibility, and others that perceive a real risk of contracting it [28].

Similar to our results, the lack of personal risk perception also varied among healthcare professionals in Singapore. The main reasons given by subjects for not considering themselves vulnerable to vaccine-preventable diseases were perceptions of high immunity, not working directly with infected patients, and considering themselves too young to be at risk for infectious diseases [33].

Concerning hesitancy regarding the specific vaccines, vaccination against hepatitis B and MMR revealed very low rates of complete schedules. Similar to our findings, the prevalence of vaccine hesitancy for hepatitis B was high among Italian health professionals. The authors of research carried out in Italy attributed the results to the memory bias related to the number of injections the subjects had received and to the impact of the vaccine controversy concerning multiple sclerosis, in which the aluminum adjuvants present in the vaccines were related to Alzheimer’s disease in the 1990s [19].

Regarding the vaccine against measles, mumps, and rubella, it is likely that, although they were not vaccinated, the workers acquired natural immunity by having contracted one or more of the three diseases when they were children. However, it is known that only the determination of a positive serological titer can confirm acquired immunity for such diseases. The prevalence of hesitancy is worrying, as such diseases are considered highly contagious [31,34], with the current risk of the measles virus in circulation in the Brazilian territory.

The prevalence of vaccine hesitancy for the analyzed vaccines was remarkably high among health agents and administrative and support staff. One hypothesis to explain this finding is the low-risk perception regarding such diseases, generally considered as childhood diseases with low incidence nowadays. Studies conducted in other health environments have shown great differences in vaccination attitudes between different professional categories, despite their proven vulnerability to infectious diseases. The study by Edge et al. [35] found evidence that institutional norms strongly influence the establishment of behaviors considered “standard”. The study, conducted in England, showed that early career physicians probably reproduce the behaviors of senior physicians [35].

The results confirm the role of social norms, of culturally accepted behaviors, as behavior inducers. Thus, when outstanding supervisors or public managers recognize and encourage specific behavior, the workers’ probability of taking and executing this behavior increases. Vaccination of superiors or immediate supervisors increases vaccine acceptance among health workers [36]. An intervention study in Spain showed that a high level of institutional support, based on public and personal commitment to vaccinate, improved vaccination rates for different professional categories. The main adopted strategies to increase vaccination coverage in the workplace were: weekly educational messages sent by e-mail, raffles of gifts for vaccinated health professionals, and an internet page with photos of vaccinated workers, giving visibility to the topic [37].

Among professionals, the association of the male sex with vaccine hesitancy was evident. Gomes et al. [38] corroborate the discussion of this result when they argue that men use health services less than women. The low demand for health services may be due to a lower concern for their health, a situation supported by the belief, socially produced and reproduced, that as men are strong, they do not need regular preventive care. The authors argue that the use of health services by men usually occurs when a disease is identified [38]. A study carried out in Bahia found that men devalue preventive care and did not recognize or adopt health prevention strategies and actions such as immunization [39]. Although the study was not conducted with health workers, it underlines the need to strengthen gender approaches in preventive practices.

In addition to gender issues, a lack of information about vaccines and campaigns or a low quality of information provided by campaigns was associated with greater vaccine hesitancy among professionals who did not work in care provision. Awareness-raising campaigns must be promoted to this group, in the same way that they are promoted to the general public, in view of this category’s hesitant attitudes towards vaccination and taking their responsibilities into account [40].

In the case of vaccines, we face a curious paradox: vaccination ends up being a “victim of its success”. The vaccine prevents diseases; when vaccine-preventable diseases are not circulating, there is a tendency to reduce the population’s demand for vaccination services, reducing vaccination coverage and producing new susceptibility. Thus, specific vaccination campaigns for different social and age groups, including health workers, must be permanent and widely publicized, even when diseases are under control. The lack of incentive to vaccinate implies the reduction of vaccine coverage and the possibility of outbreaks and epidemics, which generates a high demand for services and the resurgence of diseases already controlled.

Distance to vaccination clinics and units and financial issues have been frequently cited as barriers to access to vaccines [36]. However, our study did not find these factors associated with vaccine hesitancy. This finding might be understandable since most participants work in primary health care, where vaccines are widely available for free.

Corroborating other studies published in different countries, some factors were associated with vaccine hesitancy in the group investigated here: male sex [41], worse income conditions [42], and lack of trust in the government [43,44]. All of them are important elements that reflect beliefs and attitudes related to vaccine hesitancy.

It is paradoxical to think that, although health professionals are considered the most trusted source of vaccine-related information by the public, some of them are losing confidence in vaccines. Thus, not feeling safe to receive new vaccines was associated with vaccine hesitancy. Although hesitancy remains lower in this group compared to the general population, the analysis of confidence in this group needs attention [18]. As shown in this study, the sources of concern for vaccines among health professionals are very similar to those of the general population. Due to this, the concerns and barriers of workers themselves need to gain visibility and be better understood.

Medium complexity care workers were significantly more hesitant about vaccination than those working in primary care. According to the findings reported by Czajka et al. [40] in a study carried out in Poland, individuals who were not informed about the available vaccines were twice as likely to be distrustful of vaccines [40]. The lack of availability of the vaccine in the workplace may be a factor associated with greater hesitancy among these workers. Thus, moving to a primary care vaccination unit is necessary for workers in the medium complexity services. Considering the high workload usually performed by these workers, this may be a barrier to vaccination. This result reinforces the demand for vaccination campaigns for this group with the expanded offer of vaccination sites.

Harrison et al. [30] identified that health professionals sufficiently informed about vaccine recommendations and possible adverse events are more likely to be vaccinated. However, in this study, vaccine hesitancy was lower among those who did not trust the information about vaccinations provided by professionals. Two hypotheses may help explain this controversial result. The first hypothesis stems from the specific characteristics of the studied group—health workers. In this case, as it is a group with special health training, workers may prefer to trust their own judgment on the matter instead of relying on information from other professionals. This low trust in other professionals’ information is plausible in a social context of widespread dissemination of false information. The second hypothesis is that the item of the questionnaire (“Do you trust the information that professionals provide about vaccination”) may not have been well understood. Respondents may have interpreted that the question concerned information based on personal or biased opinions (not scientifically based knowledge). This aspect, therefore, needs to be further explored for a better understanding and, eventually, the question is written should be changed to avoid misunderstandings.

Unexpectedly, sharing information related to vaccination on social media was not associated with vaccine hesitancy. Likely, people with in-depth knowledge about vaccination, such as health professionals, have more scientifically based arguments. They may feel safe and confident in sharing information and publicizing their positions. Providing accurate and precise information can have a more significant weight in this group as they are health professionals, generating greater caution in the transfer of information or the broader dissemination of ideas. On the other hand, people who did not feel adequately and satisfactorily prepared, did not have sufficient knowledge, or have uncertainties about vaccination, are also likely to avoid sharing information on social media. Additionally, Brazil has a strong tradition of vaccinating the population [23], and vaccination is a relevant issue in health professionals’ education and training programs. Therefore, workers who are hesitant to vaccinate may feel uncomfortable assuming their hesitation and thus, choose not to share information related to the vaccine.

This study produced a diagnosis of vaccine hesitancy among health workers in primary care and medium complexity services. The results obtained can help manage health services, directing specific measures to increase vaccination which can raise the levels of biosafety at work, protect workers, and reduce potential contamination and infectious diseases. However, the study also has limitations and weaknesses. Cross-sectional studies are not capable of testing nor confirming causal relationships. Our findings are based on self-reported data. Therefore, the possibility of memory bias cannot be ruled out.

Recently, the 5C model of vaccine hesitancy determinants was proposed [45], expanding the 3C model used as the theoretical reference for this study [13]. This new model incorporates the dimensions of “calculation” (engagement in the search for information) and “collective responsibility” (willingness to protect others). Unfortunately, it was not possible to explore the potential role of these two new dimensions in our study since the period of planning our study took place simultaneously with the proposal of this new model, making the timely process of cultural adaptation of the scale unfeasible.

## 7. Conclusions

It is not our intention here to use labels that place hesitant workers in antivaccination categories. Those who do not accept vaccines, who procrastinate, or do not complete their vaccination schedules should be scrutinized. Understanding the determinant factors in this group enables adequate, goal-directed education.

In this study, we sought to investigate “vaccine hesitancy” as a concept that does not aim to reinforce a dichotomy or polarization between those against and those in favor of vaccines. Different factors contribute to vaccine hesitancy. Primary care workers presented lower hesitancy compared to medium complexity care workers. In addition, the professionals who did not work as care providers (such as administrative and support staff), even when assigned to primary care, also had different ways of accessing vaccines and information about them; consequently, they had different vaccine hesitancy profiles. Therefore, we suggest that some subjective elements be more deeply explored in qualitative studies to better understand the specific contexts and behaviors that give rise to vaccine hesitancy in different occupations.

Given the dynamic nature of vaccine hesitancy, it is necessary to monitor the vaccination status of these workers continually. Research that reveals little hesitation in one year may display a different outcome in the following year, mainly if we consider the COVID-19 pandemic, which has spotlighted the importance of vaccination. These trends need to be monitored. Additionally, qualitative research can provide information about the context of the influences that may contribute to hesitancy. Perception of the importance of vaccines for oneself and the determinants that influence the process of decision making and risk perception have not been well documented so far.

Vaccine hesitancy produces low-vaccination rates in SUS health workers. Therefore, further studies about motivators and barriers to vaccination should be conducted in specific occupational groups using the guidelines provided by the World Health Organization. 

Strategies to enhance confidence in vaccination among health workers should consider differences in occupations and their work settings. Improving vaccination content in training and continuing education activities and facilitating access to onsite vaccination at the workplace are crucial elements to reduce vaccine hesitancy among health workers.

## Figures and Tables

**Figure 1 vaccines-10-02013-f001:**
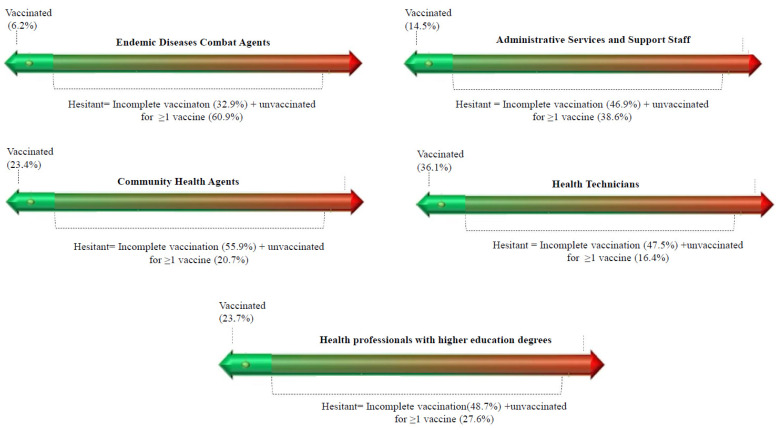
Prevalence of complete vaccination schedule, incomplete vaccination schedule and nonvaccination for at least one immunizer according to occupational categories in Bahia, Brazil, in 2020.

**Table 1 vaccines-10-02013-t001:** Sociodemographic and occupational characteristics of the health workers in Bahia, Brazil in 2020.

	Total *	%
Gender		
Female	375	82.78
Male	78	17.22
Age (years)		
21–39	198	44.80
40+	244	55.20
Skin color/Race		
White and others	72	16.36
Brown	222	50.45
Black	146	33.18
Education		
College and graduate studies	220	50.00
Up to high school	220	50.00
Lives with partner		
Yes	272	60.85
No	175	39.15
Income		
>2 minimum wages	121	32.35
Up to two minimum wages	253	67.65
Employment relationship		
Permanent	308	69.06
Temporary	138	30.94
Length of time in the profession (years)		
Up to 10	271	61.87
>10	167	38.13
Contact with biological material at work		
Never	201	44.87
Rarely	53	11.83
Sometimes	92	20.54
Always	102	22.77
Health service’s level of care		
Primary care services	352	77.70
Medium complexity care services	101	22.30
Occupation		
Community health agents	111	25.00
Endemic disease combat agents	64	14.41
Administrative or support services	132	29.73
Health technicians	61	13.74
Professionals with a higher education degree	76	17.12

* Total may vary due to missing values.

**Table 2 vaccines-10-02013-t002:** Vaccination influences, attitudes and experiences among health workers in Bahia, Brazil, in 2020.

	Total *	%
Do you share information about vaccination on social media?		
yes	259	57.30
no	193	42.70
Has anything ever happened in your life or community that made you stop believing in vaccines?		
no	421	92.94
yes	32	7.06
Do you believe that the government provides the best vaccine on the market?		
yes	293	70.60
no	122	29.40
Do you trust the industry that produces your vaccine?		
yes	342	82.61
no	72	17.39
Do you trust the information that professionals provide about vaccination?		
yes	406	90.22
no	44	9.78
Do you feel safe receiving new vaccines?		
yes	367	81.37
no	84	18.63
Does the distance to vaccination clinics or units prevent you from being vaccinated?		
no	402	96.17
yes	16	3.83
In vaccination campaigns, do you receive sufficient information to decide on being vaccinated?		
yes	367	81.56
no	83	18.44
Within your family, friends, and community circles, do people usually seek vaccination?		
yes	410	91.11
no	40	8.89
Do you feel sufficiently informed about the risks and benefits of vaccination?		
yes	350	77.26
no	103	22.74
Have you ever decided not to be vaccinated due to a lack of financial means?		
no	406	89.82
yes	46	10.18
Do you feel afraid when you are about to receive a vaccine?		
no	280	62.08
yes	171	37.92
Do you consider that the vaccination schedule is sufficiently flexible to comply with?		
yes	377	84.72
no	68	15.28
Have you ever had reactions to any vaccines?		
no	317	70.29
yes	134	29.71
Do you feel well treated by the professionals who administer vaccines?		
yes	408	91.07
no	40	8.93

* Total may vary due to missing values.

**Table 3 vaccines-10-02013-t003:** Frequency of vaccination against hepatitis B, measles, mumps, and rubella, and diphtheria and tetanus among health workers in Bahia, Brazil, in 2020.

	Total *	%
Hepatitis B vaccination schedule		
Complete	258	57.85
Incomplete	124	27.80
Unvaccinated	64	14.35
Measles–Mumps–Rubella vaccination schedule		
Complete	153	34.54
Incomplete	179	40.41
Unvaccinated	111	25.06
Diphtheria and tetanus vaccination schedule		
Complete	216	49.09
Incomplete	172	39.09
Unvaccinated	52	11.82
Vaccine hesitancy		
Complete vaccination schedule for all vaccines	91	20.45
Incomplete vaccination schedule for at least one vaccine	354	79.55

* Total may vary due to missing values.

**Table 4 vaccines-10-02013-t004:** Association between sociodemographic characteristics, occupational characteristics, vaccination-related beliefs, and vaccine hesitancy in health workers in Bahia, Brazil, in 2020.

	N	P (%)	PR (95% CI)
Gender *			
Female	367	77.1	1.00
Male	78	91.0	1.18 (1.07–1.29)
Age (years)			
21–39	196	78.0	1.00
40+	239	81.1	1.03 (0.94–1.14)
Skin color or Race			
Nonblack	288	78.4	1.00
Black	144	80.5	1.02 (0.92–1.13)
Education			
College and graduate studies	216	79.1	1.00
Up to high school	216	79.1	1.00 (0.90–1.10)
Lives with partner *			
Yes	266	77.4	1.00
No	173	82.0	1.05 (0.96–1.16)
Income *			
>2 minimum wages	120	81.2	1.00
Up to two minimum wages	249	80.7	1.08 (0.96–1.22)
Employment relationship			
Permanent	304	81.2	1.00
Temporary	134	76.8	0.94 (0.84–1.05)
Length of time in the profession (years)			
Up to 10	265	78.8	1.00
>10	166	78.9	1.00 (0.90–1.10)
Contact with biological material at work			
No	195	79.4	1.00
Yes	245	79.1	0.99 (0.90–1.09)
Health service’s level of care *			
Primary care services	346	76.8	1.00
Medium complexity care services	99	88.8	1.15 (1.05–1.26)
Occupation *			
Community health agents	109	76.1	1.19 (0.95–1.47)
Endemic disease combat agents	64	93.7	1.46 (1.20–1.78)
Administrative and support services	126	84.9	1.32 (1.08–1.62)
Health technicians	61	63.9	1.00
Professionals with higher education degree	76	76.3	1.19 (0.95–1.49)
Do you share information about vaccination on social media? *			
Yes	253	72.3	1.00
No	191	89.0	1.23 (1.12–1.34)
Has anything ever happened in your life or community that made you stop believing in vaccines?			
No	414	79.2	1.00
Yes	31	83.8	1.05 (0.90–1.24)
Do you believe that the government provides the best vaccine on the market? *			
Yes	287	77.0	1.00
No	121	85.9	1.11 (1.01–1.22)
Do you trust the industry that produces your vaccine? *			
Yes	336	78.2	1.00
No	71	87.3	1.11 (1.00–1.23)
Do you trust the information that professionals provide about vaccination? *			
Yes	400	78.7	1.00
No	43	86.0	1.09 (0.95–1.24)
Do you feel safe receiving new vaccines? *			
Yes	361	76.7	1.00
No	82	91.4	1.19 (1.09–1.30)
Does the distance to vaccination clinics or units prevent you from being vaccinated? *			
No	395	78.9	1.00
Yes	16	93.7	1.18 (1.03–1.36)
In vaccination campaigns, do you receive sufficient information to decide on being vaccinated? *			
Yes	361	77.5	1.00
No	82	87.8	1.13 (1.02–1.24)
Within your family, friends, and community circles, do people usually seek vaccination?			
Yes	403	79.4	1.00
No	40	80.0	1.00 (0.85–1.18)
Do you feel sufficiently informed about the risks and benefits of vaccination? *			
Yes	344	78.2	1.00
No	101	84.1	1.07 (0.97–1.19)
Have you ever decided not to be vaccinated due to a lack of financial means? *			
No	398	78.3	1.00
Yes	46	89.1	1.13 (1.01–1.27)
Do you feel afraid when you are about to receive a vaccine? ^§^			
No	275	79.2	1.00
Yes	168	79.7	1.00 (0.91–1.10)
Do you consider that the vaccination schedule is sufficiently flexible to comply with? *			
Yes	370	76.4	1.00
No	67	94.0	1.22 (1.13–1.33)
Have you ever had reactions to any vaccines? ^§^			
No	132	79.5	1.00
Yes	312	79.4	0.99 (0.90–1.10)
Do you feel well treated by the professionals who administer vaccines? *			
Yes	400	78.5	1.00
No	40	87.5	1.11 (0.98–1.26)

* Variables selected for the multivariate model (*p*-value ≤ 0.25). ^§^ Variables forced into the multivariate model.

**Table 5 vaccines-10-02013-t005:** Factors associated with vaccine hesitancy among health workers obtained in multivariate analysis in Bahia, Brazil, in 2020.

	PR (95% CI)
Health service’s level of care	
Primary care services	1.00
Medium complexity care services	1.21 (1.07–1.36)
Occupation	
Community health agents	1.25 (1.01–1.55)
Endemic disease combat agents	1.42 (1.16–1.75)
Administrative and support services	1.22 (1.00–1.48)
Health technicians	1.00
Professionals with higher education degree	1.11 (0.89–1.39)
Do you share information about vaccination on social media?	
Yes	1.00
No	1.16 (1.05–1.28)
Do you trust the information that professionals provide about vaccination?	
Yes	1.00
No	0.86 (0.75–0.99)
Do you feel safe receiving new vaccines?	
Yes	1.00
No	1.16 (1.06–1.28)

Controlled by variables in the model.

## Data Availability

The data presented in this study are available on request from the corresponding author. The data are not publicly available due to confidentiality issues with the individuals surveyed.

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
