# Peer review of "Beliefs and Sociodemographic and Occupational Factors Associated with Vaccine Hesitancy among Health Workers"

_vaccines, 2022, doi:10.3390/vaccines10122013_

Round 1

Reviewer 1 Report

Vaccines  -2010173 - Review Request

Beliefs, Sociodemographic and occupational factors associated with vaccine hesitancy among health workers.

Dear authors,

thank you for the submission of your manuscript. Attached you will find my comments:

The aim of this study is to investigate beliefs, socio-demographic and occupational factors associated with vaccine hesitancy among health workers in Bahia, Brazil. The authors used data from the research project "Vigilância e Monitoramento de Doenças Infecciosas entre Trabalhadores e Trabalhadoras do Setor Saúde" and could examine n=453 health workers using questionnaires. The variable vaccination hesitancy was operationalized by the self-reported vaccination status against hepatitis B, measles, mumps and rubella, and diphtheria and tetanus.

Using bivariate and multivariate analyses, the authors were able to identify a significant association between vaccination hesitancy and the following variables: working in secondary health care services, working as endemic diseases combat agent, not sharing information about vaccines on the social media, distrusting information about vaccination and not feeling safe about receiving new vaccines.

The study focusses a very important topic and gives valuable hints that might be useful to develop measures, which could prevent vaccine hesitancy in different groups of health workers.

I would recommend: ‘revision’.

 Abstract

Introduction (line 11-15): please use shorter and less complex sentences to describe general issues such as low vaccination rates in different countries or the role of health care workers.

Manuscript

Introduction:

Line 48-51: please indicate that besides confusing or false information about vaccines, there are some other reasons for vaccine hesitancy (e.g. reduced health literacy, organizational constraints, reduced feeling of social responsibility and many more).

Line 54-55: it is mentioned, that there is “(…) little known about the health workers`general vaccination status.” Please add a sentence or two, whether the vaccination status is recorded and documented by the clinic administration or the internal medical service of healthcare institutions. At least for the working environment in the healthcare system, this is standard in many European countries. Furthermore, there seems to be some literature concerning this issue (e.g. reference number 12)?

Line 66-70: you should explain the different job titles and their functions of the analyzed health workers in short, as they are not known in other health care systems. For example: what is meant by “medium complexity care services”, what kind of occupations or services are included? Which occupations belong to the Brazilian primary care system? What is a “community health agent” or an “endemic diseases combat agent” doing and in which context? Are these health workers treating patients directly? What kind of patients in which situation?

Furthermore, it is important to understand the accessibility in terms of vaccinations for these different types of health workers. Are all of the analyzed vaccinations available on a low-threshold basis? Do the health workers have to pay for these vaccinations? Do they have a reminder system or regulatory consultation, which pays attention to the individual vaccination status of each health worker? Do they have a vaccination book (or comparable document) or a digital vaccination register for each citizen in Bahia/Brazil?

Line 72 – 90: The term "vaccine hesitancy" was explained. However, how do you differentiate between “hesitancy” and other reasons for an incomplete vaccination status e.g. forgetting to vaccinate? Within the section “Methods” you categorize every health worker to be “hesitant” if he or she is not fully vaccinated.

Methods:

Line 107: please give a short description of the study (up to three sentences would be enough). The website should be cited as a reference.

Line 109: please indicate if this was a mono-centre or mulitcentre-study. From my point of view, it is hard to assess the representativeness of your survey.

Line 127-129: "(…)who were eligible for the study". Please define the inclusion and exlusion criteria of your population.

Line 133-136: Why did you choose different forms of questioning (with and without a trained interviewer)? Please explain possible reasons: e.g. language barriers, health literacy? How did you avoid a bias due to these different ways of data collection?

Line 137: Did you use validated questionnaires? Did you check the quality criteria of the applied questionnaires?

Line 150-151: you categorize every health worker to be “hesitant” if he or she is not fully vaccinated. There might be some more reasons for being not fully vaccinated than “hesitancy” (e.g. missing a shot due to missing information about the importance of follow-up shots or having been contracted already by a disease like measles).

Line 173: Why did you choose p≤0.25?

Results

Line 206: "...but 29.7% reported having a history of reactions concerning some vaccine (Table 2)". In table 2 the percentiles are shown vice versa: "Have you ever had reactions to any vaccines?" "Yes": 70.29% /No": 29.71%” Please correct the data in the text or within table 2.

Discussion

Line 256-259: how do you know about the formal degree of each participant? Did you ask for that in the survey? Is this the occupational group “administrative/support service”? Or participants with an educational degree “up to high school”?

Line 280: Comparable to our comments above (referring to line 150): Please reconsider using the term "vaccine hesitancy" for your study. Rather, it is a question of reasons for insufficient vaccination rates. However, these reasons can vary and are not exclusively of a psychological nature. That was partly considered in this section of the discussion.

Line 287: Where exactly does this hypothesis come from? Please justify

Line 312: Your cited study (Number 40) should be placed in context to assess the extent of comparability to your results. Where the participants from general population or also health workers?

Thank you for your valuable contribution!

Best regards 

Author Response

Response to Reviewer 1 - Comments

Abstract

Introduction (line 11-15): please use shorter and less complex sentences to describe general issues such as low vaccination rates in different countries or the role of health care workers.

Response: We changed the sentences as suggested, reducing the number of words from 54 to 38. The new text reads as follows:

“Vaccine hesitancy has been implicated in the low vaccination coverage in several countries. Knowledge about vaccine hesitancy predictors in health workers is essential because they play a central role in communication about the importance and safety of vaccines.”

Manuscript

Introduction:

Line 48-51: please indicate that besides confusing or false information about vaccines, there are some other reasons for vaccine hesitancy (e.g. reduced health literacy, organizational constraints, reduced feeling of social responsibility and many more).

Response: We changed the text to accept the suggestions, now it reads:

“Even in the pandemic context, vaccine hesitancy among adults have been frequently registered due to misinformation about the efficacy and safety of vaccines, reduced health literacy, organizational constraints, and reduced feelings of social responsibility [14,15].”

Line 54-55: it is mentioned, that there is “(…) little known about the health workers`general vaccination status.” Please add a sentence or two, whether the vaccination status is recorded and documented by the clinic administration or the internal medical service of healthcare institutions. At least for the working environment in the healthcare system, this is standard in many European countries. Furthermore, there seems to be some literature concerning this issue (e.g. reference number 12)?

Response: We included a sentence indicating that, in Brazil, monitoring of vaccination status in the health care system is not enforced. The text now reads:

“In Brazil, the vaccination schedule for health workers recommended by the Ministry of Health includes vaccines against hepatitis B, diphtheria and tetanus (dT), measles, mumps, and rubella (MMR) [16]. Despite these recommendations and the free-of-charge access to vaccines, a complete vaccination schedule as low as 38.5% has been described among health workers [12]. Among the possible challenges to reaching high vaccination rates in Brazilian health workers is the availability of vaccination rooms near their workplaces. Since vaccination rooms are mainly located in primary care units, workers outside these settings are likely to have fewer opportunities for vaccination. In addition, most healthcare services have no standardized reminder system for individual vaccination assessment. That is, there is no systematic monitoring or surveillance of the vaccination status of health workers.”

Line 66-70: you should explain the different job titles and their functions of the analyzed health workers in short, as they are not known in other health care systems. For example: what is meant by “medium complexity care services”, what kind of occupations or services are included? Which occupations belong to the Brazilian primary care system? What is a “community health agent” or an “endemic diseases combat agent” doing and in which context? Are these health workers treating patients directly? What kind of patients in which situation?

Response: Although we agree with the reviewer, this specific paragraph was shaped to reinforce the importance of health workers in the success of a vaccination program. Detailed information regarding jobs and functions and the organization of health services in Brazil is described in the second paragraph of the Methods section. We rewrote this paragraph for clarity and added the information required in the Methods section of the manuscript. Now the text reads:

“This survey included workers in effective professional practice who worked in Primary care and Medium Complexity services in urban and rural areas of a municipality in Bahia, Northeast Brazil, in 2019. Professional in primary care services mainly work in clinical care, surveillance, and health promotion activities. Among them are the so-called “community health agents” and the “endemic diseases combat agents”. The first develops health promotion and disease preventive actions, focusing on health education activities in households and communities. The second, also works outside the health unit inspecting houses, deposits, vacant lots and commercial establishments with the aim of preventing and controlling infectious diseases.

Medium-complexity health services are not directly related to the vaccination procedures, and include health workers from different backgrounds developing a variety of activities such as specialized clinical and psychosocial care, emergency care, occupational health monitoring, testing and counseling for infectious diseases, and health management [12].

All the workers who were effectively working, enrolled in the National Register of Health Facilities (CNES), were eligible for the study. Nominal lists of these workers, provided by the municipal health department, were checked at the workplace during the initial visits to the health services.”

Furthermore, it is important to understand the accessibility in terms of vaccinations for these different types of health workers. Are all of the analyzed vaccinations available on a low-threshold basis? Do the health workers have to pay for these vaccinations? Do they have a reminder system or regulatory consultation, which pays attention to the individual vaccination status of each health worker? Do they have a vaccination book (or comparable document) or a digital vaccination register for each citizen in Bahia/Brazil?

Response: All workers can access the recommended vaccines free of charge. However, since vaccination rooms are mainly located in primary care units, workers in these settings are likely to have more opportunities for vaccination. All Brazilian citizens should have a printed vaccination book, but the existing digital vaccination system still needs improvement concerning its coverage and completeness. Most healthcare services have no standardized reminder system for individual vaccination assessment. As described above, such information was included in a paragraph in the Introduction section:

“In Brazil, the vaccination schedule for health workers recommended by the Ministry of Health includes vaccines against hepatitis B, diphtheria and tetanus (dT), measles, mumps, and rubella (MMR) [16]. Despite these recommendations and the free-of-charge access to vaccines, a complete vaccination schedule as low as 38.5% has been described among health workers [12]. Among the possible challenges to reaching high vaccination rates in Brazilian health workers is the availability of vaccination rooms near their workplaces. Since vaccination rooms are mainly located in primary care units, workers outside these settings are likely to have fewer opportunities for vaccination. In addition, most healthcare services have no standardized reminder system for individual vaccination assessment. That is there is no systematic monitoring or surveillance of the vaccination status of health workers in the investigated services.”

Line 72 – 90: The term "vaccine hesitancy" was explained. However, how do you differentiate between “hesitancy” and other reasons for an incomplete vaccination status e.g. forgetting to vaccinate? Within the section “Methods” you categorize every health worker to be “hesitant” if he or she is not fully vaccinated.

Response: We agree with the reviewer that defining vaccine hesitancy in practice is a challenge because vaccine hesitancy is complex, involves specific aspects of each context, and is influenced by many factors. It might encompass a wide range of people who differ from those who refuse all vaccinations and those who have no doubts about getting a vaccine. Considering such heterogeneity, the SAGE Working Group on Vaccine Hesitancy defines vaccine hesitancy as any delay in acceptance or refusal of vaccination despite availability of vaccination services. Therefore, we understand that considering as vaccine hesitant those who have stopped receiving vaccines for any reason is consistent with the SAGE definition. (Reference 13 - MacDonald NE, Eskola J, Liang X, Chaudhuri M, Dube E, Gellin B, et al. Vaccine hesitancy: Definition, scope and determinants. Vaccine 2015;33(34):4161–4. https://doi.org/10.1016/j.vaccine.2015.04.036.)

Methods:

Line 107: please give a short description of the study (up to three sentences would be enough).

Response: We added the following text in the Methods section:

“The primary purpose of the research program is to develop knowledge and social technologies to structure public policies for improving the health and working conditions of health workers in Bahia, Brazil. Specifically, we aim to estimate the prevalence of infectious diseases and vaccine uptake among health workers, and to evaluate preventive measures and propose a long-term monitoring system for health workers.”

Line 109: please indicate if this was a mono-centre or mulitcentre-study. From my point of view, it is hard to assess the representativeness of your survey.

Response: This study was developed among health workers from various health services from one city in Bahia, Brazil. We indicate this information in the text.

“This survey included workers in effective professional practice who worked in Primary care and Medium Complexity services in urban and rural areas of a municipality in Bahia, Northeast Brazil, in 2019.”

Line 127-129: "(…) who were eligible for the study". Please define the inclusion and exclusion criteria of your population.

Response: All health workers, in effective professional practice, from municipal primary care and medium complexity health units were included in the study:

“This survey included workers in effective professional practice who worked in Primary care and Medium Complexity services in urban and rural areas of a municipality in Bahia, Northeast Brazil, in 2019.”

“All the workers who were effectively working, enrolled in the National Register of Health Facilities (CNES), were eligible for the study. Nominal lists of these workers, provided by the Municipal Health Department, were checked at the workplace during the initial visits to the health services.”

Line 133-136: Why did you choose different forms of questioning (with and without a trained interviewer)? Please explain possible reasons: e.g. language barriers, health literacy? How did you avoid a bias due to these different ways of data collection?

Response: The decision to choose different form of questioning was based on language barriers. A pilot-study was used to standardize both procedures to avoid information bias. The text now reads as follows:

“Data were collected using two procedures. For professionals with higher education degrees, we used a structured, self-administered questionnaire. Alternatively, considering possible language barriers and different levels of education, a face-to-face questionnaire was administered by trained interviewers for professionals with lower schooling levels. A pilot study was conducted to test and standardize procedures to minimize information bias.”

Line 137: Did you use validated questionnaires? Did you check the quality criteria of the applied questionnaires?

Response: This is a multidimensional questionnaire in which some questions built specifically for this research and others, concerning vaccination-related beliefs, extracted from a validated questionnaire (Reference number 28, Neves CR, Codeço CT, Luz PM, Garcia LMT. Preditores de aceitação da vacina contra influenza: tradução para o português e validação de um questionário. Cad Saúde Pública 2020; 36 Suppl 2:e00211518.)

We included the following text in the methods section to acknowledge this recommendation:

“The multidimensional questionnaire used for data collection included questions built specifically for this research, based on a literature review focusing on work and health conditions in the healthcare sector, and questions concerning vaccination-related beliefs, extracted from a validated questionnaire [28].”

Line 150-151: you categorize every health worker to be “hesitant” if he or she is not fully vaccinated. There might be some more reasons for being not fully vaccinated than “hesitancy” (e.g. missing a shot due to missing information about the importance of follow-up shots or having been contracted already by a disease like measles).

Response: As we mentioned above, defining vaccine hesitancy in practice is challenging, because vaccine hesitancy is complex, involves specific aspects of each context, and is influenced by many factors. It might encompass a wide range of people who differ from those who refuse all vaccinations and those who have no doubts about getting a vaccine. Considering such heterogeneity, the SAGE Working Group on Vaccine Hesitancy defines vaccine hesitancy as any delay in acceptance or refusal of vaccination despite availability of vaccination services. Therefore, we understand that considering as vaccine hesitant those who have stopped receiving vaccines for any reason is consistent with the SAGE definition. (Reference 13 - MacDonald NE, Eskola J, Liang X, Chaudhuri M, Dube E, Gellin B, et al. Vaccine hesitancy: Definition, scope and determinants. Vaccine 2015;33(34):4161–4. https://doi.org/10.1016/j.vaccine.2015.04.036.)

Line 173: Why did you choose p≤0.25?

Response: The p≤0.25 threshold for including variables in the multivariate model has been suggested elsewhere as an appropriate threshold and is widely used in epidemiologic research. (Reference 29 - Hosmer DW, Lemeshow S. Applied Logistic Regression. 2th ed. New York: John Wiley & Sons; 2000)

Results

Line 206: "...but 29.7% reported having a history of reactions concerning some vaccine (Table 2)". In table 2 the percentiles are shown vice versa: "Have you ever had reactions to any vaccines?" "Yes": 70.29% /No": 29.71%” Please correct the data in the text or within table 2.

Response: Thank you for detecting this mistake. The text contains the right information; we corrected it in the table. "Have you ever had reactions to any vaccines?" "No": 70.29% /Yes": 20.71%”

Discussion

Line 256-259: how do you know about the formal degree of each participant? Did you ask for that in the survey? Is this the occupational group “administrative/support service”? Or participants with an educational degree “up to high school”?

Response: Health service workers may have specific training in undergraduate health courses or not. We asked for that in the survey, so it was possible to establish such differentiation.

Line 280: Comparable to our comments above (referring to line 150): Please reconsider using the term "vaccine hesitancy" for your study. Rather, it is a question of reasons for insufficient vaccination rates. However, these reasons can vary and are not exclusively of a psychological nature. That was partly considered in this section of the discussion.

Response: We understand that using the term vaccine hesitancy in our study is consistent with the definition used by the SAGE Working Group on Vaccine Hesitancy (vaccine hesitancy is defined as any delay in acceptance or refusal of vaccination despite the availability of vaccination services). The SAGE definition considers vaccine hesitancy as a complex and context-specific phenomenon influenced by many factors, not only those of psychological nature. (Reference 13 - MacDonald NE, Eskola J, Liang X, Chaudhuri M, Dube E, Gellin B, et al. Vaccine hesitancy: Definition, scope and determinants. Vaccine 2015;33(34):4161–4. https://doi.org/10.1016/j.vaccine.2015.04.036.)

Line 287: Where exactly does this hypothesis come from? Please justify

Response. Thanks for the comment. The hypothesis concerns the low-risk perception since vaccine-preventable diseases are viewed as childhood diseases with low incidence nowadays. We changed the phrase; now it reads:

“One hypothesis to explain this finding is the low-risk perception regarding such diseases, generally considered as childhood diseases with low incidence nowadays.

Line 312: Your cited study (Number 40) should be placed in context to assess the extent of comparability to your results. Where the participants from general population or also health workers?

Response: The text has been reorganized to ensure better clarity:

“A study carried out in Bahia found that men devalue preventive care and do not recognize or adopt health prevention strategies and actions such as immunization [40]. Although the study was not conducted with health workers, it underlines the need to strengthen gender approaches in preventive practices.”

Thank you for your valuable contribution!

Best regards.

Reviewer 2 Report

Abstract: The abstract is extremely long and not appealing. The introduction and conclusion section can be condensed and improved.

The authors states “There is a gap in knowledge about vaccine hesitancy predictors in different occupational groups, particularly health workers”. This statement is incomplete and misleading in abstract. In the introduction, the authors acknowledges that there are some studies examining the factors influencing healthcare workers’ (HCWs) vaccine hesitancy.

Keywords: Please consider adding a few more keywords.

Introduction:       The authors state “little is known about the health workers’ general vaccination status”(lien 55). Line 93 “Few studies approach vaccine status in general [26]”. Please discuss the findings of these studies in greater detail and then identify the research gaps and position your study. The author should discuss in more detail how this investigation affects the field, and what it adds to existing research

·       You could define vaccine hesitancy earlier in the Introduction section.

·       Cite articles from 2022 if available

Materials & Methods:       This section is lengthy and does not flow well as it touches multiple topics but ideas are not connected well. The content can be reorganized under different subheadings such as sample and data collection procedure, measure, data analysis………

·       Provide a citation supporting your approach for calculating sample size

·       Line 137, “The questionnaire used for data collection was developed based on a literature review that focused on work and health conditions in the healthcare sector”. This is a vague statement. Please further elaborate. If the survey items/scales were borrowed from previous studies, please provide the sources of key measures.

Limitations and Agendas for Future Research:       Consider expanding study limitations and offer a couple of agendas for future research

Author Response

Response to Reviewer 2 - Comments

Abstract: The abstract is extremely long and not appealing. The introduction and conclusion section can be condensed and improved.

Response: We improved the text and reduced the introduction section from 72 to 56 words, and the conclusion section from 80 to 49 words.

The authors states “There is a gap in knowledge about vaccine hesitancy predictors in different occupational groups, particularly health workers”. This statement is incomplete and misleading in abstract. In the introduction, the authors acknowledges that there are some studies examining the factors influencing healthcare workers’ (HCWs) vaccine hesitancy.

Response: We appreciate the reviewer comment. We meant to indicate that most studies are found in the international literature and focused on the specific type of health workers that provide direct care to the patient, generally not including different categories of health workers. We changed the phrase; now it reads:

“Knowledge about vaccine hesitancy predictors in health workers is essential because they play a central role in communication about the importance and safety of vaccines.”

Keywords: Please consider adding a few more keywords.

Response: Thanks for the suggestion. We added the following keywords: vaccination delay; vaccination awareness; attitude to health; community health workers; health personnel.

Introduction:    The authors state “little is known about the health workers’ general vaccination status”(lien 55). Line 93 “Few studies approach vaccine status in general [26]”. Please discuss the findings of these studies in greater detail and then identify the research gaps and position your study. The author should discuss in more detail how this investigation affects the field, and what it adds to existing research.   You could define vaccine hesitancy earlier in the Introduction section.     

Response: We defined vaccine hesitancy earlier in the introduction section. Now the definition of vaccine hesitancy is in the second paragraph. (“Vaccine hesitancy is defined as any delay in acceptance or refusal of vaccination de-spite the availability of vaccination services [13].)

Additional information was included as required:

Introduction section:

“Studies that have examined vaccination-related barriers are often limited to hospital service professionals [8]. Thus, understanding this phenomenon concerning hesitation about vaccination among health workers in the most diverse workplaces (such as in primary care services) may guide actions to expand vaccination [26]. The analysis focus on health workers is justified because these professionals play a crucial role in promoting vaccine acceptance, in addition to being at increased risk for infection [12].”

Discussion section:

“The results obtained can help manage health services, directing specific measures to increase vaccination, which can raise the levels of biosafety at work, protect workers, and reduce potential contamination and infectious diseases.”

Conclusion section:

“Given the dynamic nature of vaccine hesitancy, it is necessary to monitor the vaccination status of these workers continually. Research that reveals little hesitation in one year may display a different outcome in the following year, mainly if we consider the Covid-19 pandemic, which has spotlighted the importance of vaccination. These trends need to be monitored. Additionally, qualitative research can provide information about the context of the influences that may contribute to hesitancy. Perception of the importance of vaccines for oneself and the determinants that influence the process of decision-making and risk perception have not been well documented so far.”

“Strategies to enhance confidence in vaccination among health workers should consider differences in occupations and their working settings. Improving the content about vaccination in training and continuing education activities and facilitating access to on-site vaccination at the workplace are crucial elements to reduce vaccine hesitancy among health workers.”

Materials & Methods:     

This section is lengthy and does not flow well as it touches multiple topics but ideas are not connected well. The content can be reorganized under different subheadings such as sample and data collection procedure, measure, data analysis.

Response: We included subheading as recommended (Participants and sample, Data collection and study variables, Statistical analysis, Ethical aspects).

       Provide a citation supporting your approach for calculating sample size:

Response: We quoted reference 27 to give support for the prevalence of vaccine hesitancy, the key parameter for defining the sample size.

          Line 137, “The questionnaire used for data collection was developed based on a literature review that focused on work and health conditions in the healthcare sector”. This is a vague statement. Please further elaborate. If the survey items/scales were borrowed from previous studies, please provide the sources of key measures.

      Response: This is a multidimensional questionnaire in which some questions built specifically for this research and others, concerning vaccination-related beliefs, extracted from a validated questionnaire (Reference number 28, Neves CR, Codeço CT, Luz PM, Garcia LMT. Preditores de aceitação da vacina contra influenza: tradução para o português e validação de um questionário. Cad Saúde Pública 2020; 36 Suppl 2:e00211518.)

We included the following text in the methods section to acknowledge this recommendation:

“The multidimensional questionnaire used for data collection included questions built specifically for this research, based on a literature review focusing on work and health conditions in the healthcare sector, and questions concerning vaccination-related beliefs, extracted from a validated questionnaire [28].”         

Limitations and Agendas for Future Research:   

Consider expanding study limitations and offer a couple of agendas for future research.

Response. We expanded and improved the description of study limitations and the agenda for further research:

“The results obtained can help manage health services, directing specific measures to increase vaccination, which can raise the levels of biosafety at work, protect workers, and reduce potential contamination and infectious diseases. However, the study also has limitations and weaknesses. Cross-sectional studies are not capable of testing nor confirming causal relationships. Our findings are based on self-reported data. Therefore, the possibility of memory bias cannot be ruled out.” 

“Recently, the 5C model of vaccine hesitancy determinants was proposed [46], expanding the 3C model used as the theoretical reference for this study [13]. This new model incorporates the dimensions of “calculation” (engagement in the search for information) and “collective responsibility” (willingness to protect others). Unfortunately, it was not possible to explore the potential role of these two new dimensions in our study since the period of planning our study took place simultaneously with the proposal of this new model, making the timely process of cultural adaptation of the scale unfeasible.” (new reference [46] Betsch C, Schmid P, Heinemeier D, Korn L, Holtmann C, Böhm R. Beyond confidence: development of a measure assessing the 5C psychological antecedents of vaccination. PLoS One 2018; 13:e0208601. https://doi.org/ 10.1371/journal.pone.0208601.)”

“Therefore, we suggest that some subjective elements be more deeply explored in qualitative studies to understand better the specific contexts and behaviors that give rise to vaccine hesitancy in different occupations.”

“Given the dynamic nature of vaccine hesitancy, it is necessary to monitor the vaccination status of these workers continually. Research that reveals little hesitation in one year may display a different outcome in the following year, mainly if we consider the Covid-19 pandemic, which has spotlighted the importance of vaccination. These trends need to be monitored. Additionally, qualitative research can provide information about the context of the influences that may contribute to hesitancy. Perception of the importance of vaccines for oneself and the determinants that influence the process of decision-making and risk perception have not been well documented so far.”

“Strategies to enhance confidence in vaccination among health workers should consider differences in occupations and their working settings. Improving the content about vaccination in training and continuing education activities and facilitating access to on-site vaccination at the workplace are crucial elements to reduce vaccine hesitancy among health workers.”

Reviewer 3 Report

The authors conducted a study of vaccine hesitancy in health workers population in the state of Bahia. They found some vaccine hesitancy among health workers and discuss the reasons of this hesitancy. The manuscript is well written and the results are clearly discussed.

The authors could have defined more clearly what is vaccine hesitancy as defined by WHO in ref 22, as the "3C model" (confidence, complacency, convenience) on which their questionnaire is based.

They discussed the perception of susceptibility to disease but do not discuss the collective responsibility feeling among health worker. As discussed (line 411-413) , including these questions would have been helpful for refining the profile of health workers reluctant to vaccination.

Author Response

Response to Reviewer 3 Comments

The authors conducted a study of vaccine hesitancy in health workers population in the state of Bahia. They found some vaccine hesitancy among health workers and discuss the reasons of this hesitancy. The manuscript is well written and the results are clearly discussed.

Response: Thank you for taking the time to read our manuscript and for your feedback.

The authors could have defined more clearly what is vaccine hesitancy as defined by WHO in ref 22, as the "3C model" (confidence, complacency, convenience) on which their questionnaire is based.

Response: Thanks for the suggestion, we added a text to take care of the recommendation.

“Considered a complex behavioral phenomenon regarding its determinants, vaccine hesitancy refers to delay in acceptance or refusal of vaccination, despite availability of access to the vaccine. It can be described on a continuum that ranges from the positive extreme - those who accept all vaccines with no doubts - to the negative extreme - those who refuse everything with no doubts. The review of these models reinforced that vaccine hesitancy is complex and is not driven by a simple set of factors. Aspects such as complacency, convenience and trust should be considered [13,23].”

They discussed the perception of susceptibility to disease but do not discuss the collective responsibility feeling among health worker. As discussed (line 411-413), including these questions would have been helpful for refining the profile of health workers reluctant to vaccination.

Response: The 5C scale, published in 2018, is a psychometrically validated tool that assesses five psychological antecedents of vaccination (trust, complacency, restraints, calculation, and collective responsibility), capturing an individual's attitude and behavioral. Unfortunately, we could not incorporate this new scale into our assessment instruments because the planning period of our study took place simultaneously with the proposal of this new model, making the process of cultural adaptation of the scale unfeasible in a timely manner.

We added the following text in the discussion section to acknowledge such limitation:

“Recently, the 5C model of vaccine hesitancy determinants was proposed [46], expanding the 3C model used as the theoretical reference for this study [13]. This new model incorporates the dimensions of “calculation” (engagement in the search for information) and “collective responsibility” (willingness to protect others). Unfortunately, it was not possible to explore the potential role of these two new dimensions in our study since the period of planning our study took place simultaneously with the proposal of this new model, making the timely process of cultural adaptation of the scale unfeasible.” (new reference [46] Betsch C, Schmid P, Heinemeier D, Korn L, Holtmann C, Böhm R. Beyond confidence: development of a measure assessing the 5C psychological antecedents of vaccination. PLoS One 2018; 13:e0208601. https://doi.org/ 10.1371/journal.pone.0208601.)
